# SARS-CoV-2 at the Human–Animal Interface: Implication for Global Public Health from an African Perspective

**DOI:** 10.3390/v14112473

**Published:** 2022-11-09

**Authors:** Ebere Roseann Agusi, Valerie Allendorf, Emmanuel Aniebonam Eze, Olayinka Asala, Ismaila Shittu, Klaas Dietze, Frank Busch, Anja Globig, Clement Adebajo Meseko

**Affiliations:** 1National Veterinary Research Institute, Vom 930001, Nigeria; 2Institute of International Animal Health/One Health, Friedrich-Loeffler-Institut, 17493 Greifswald-Insel Riems, Germany; 3Department of Microbiology, University of Nigeria Nsukka, Enugu 410001, Nigeria; 4College of Veterinary Medicine, University of Minnesota, Minneapolis, MN 55455, USA

**Keywords:** SARS-CoV-2, COVID-19, zoonosis, one health, emerging infectious disease, Africa

## Abstract

The coronavirus disease 2019 (COVID-19) pandemic has become the most far-reaching public health crisis of modern times. Several efforts are underway to unravel its root cause as well as to proffer adequate preventive or inhibitive measures. Zoonotic spillover of the causative virus from an animal reservoir to the human population is being studied as the most likely event leading to the pandemic. Consequently, it is important to consider viral evolution and the process of spread within zoonotic anthropogenic transmission cycles as a global public health impact. The diverse routes of interspecies transmission of SARS-CoV-2 offer great potential for a future reservoir of pandemic viruses evolving from the current SARS-CoV-2 pandemic circulation. To mitigate possible future infectious disease outbreaks in Africa and elsewhere, there is an urgent need for adequate global surveillance, prevention, and control measures that must include a focus on known and novel emerging zoonotic pathogens through a one health approach. Human immunization efforts should be approached equally through the transfer of cutting-edge technology for vaccine manufacturing throughout the world to ensure global public health and one health.

## 1. Introduction

The magnitude of morbidity and mortality of the severe acute respiratory syndrome coronavirus 2 (SARS-CoV-2) pandemic has led to serious and far-reaching impacts on healthcare systems, societies, economies, and politics worldwide [1]. To prepare for and potentially prevent the occurrence of such an event, research was initially centered on the probable origin of the zoonotic virus and its mechanism of spillover to humans [2]. Nonetheless, the increasing reports on zoonotic transmissions from humans to animals (anthroponosis) [3] and evidence of bi-directional transmissions at the human–animal interface [4] point to potential risks in the modification, adaptation, and perpetuation of the pathogen in nature. Consequently, human–animal interfaces with the potential for spillover infections from animals to humans and vice versa pose a significant and continuous global public health threat.

As in the example of SARS-CoV-2, it is estimated that over 75% of emerging human infectious diseases have animals as the primary source [5]. Age-long close contact between humans and domesticated species enabled the early transmission and co-evolution (e.g., measles, smallpox) of the most adaptable pathogens to humans [6]. In comparison with domestic animals, opportunities for close contact between humans and wildlife are relatively rare. Yet, the recent emergence of many diseases, such as severe acute respiratory syndrome (SARS) [7], Ebola [8], monkeypox [9], or Nipah virus encephalitis [10], demonstrate the increased risk of a spillover of wildlife-sourced pathogens into the human population [11]. The continuous growth of the global population leads to increasing demand for food and natural resources [12,13]. This, in turn, has led to changes in land use, continuous human encroachment, and changes in ecosystems, resulting in overlapping habitats of wildlife, domestic animals, and humans, which are regarded as the main drivers of zoonotic pathogen emergence [14].

Key characteristics evaluated for associations with high-risk disease emergence include host plasticity [15] spatio-temporal distribution [16], and human-to-human transmissibility [17]. In addition, human practices that facilitate zoonotic and inter-human transmission act synergistically to promote viral emergence [18]. Predominantly respiratory viruses with the capacity for rapid human-to-human transmission may then succeed in broad geographic spread [18].

The key objective of this review is to assess the extent of potential anthropogenic transmission of SARS-CoV-2 to domestic and wild animals from an African perspective and its implication on viral evolution and global public health. In the derivation of information and data used for this review, a qualitative approach was adopted. Research keywords such as *zoonoses*, *SARS-CoV-2*, *emerging diseases in Africa*, and *one health* and journal articles published between the years 2001 to 2022 were mostly considered. Over 100 peer-reviewed and pre-print articles as well as statistical data from governmental websites, published until March 2022, were the sources adopted for this work.

## 2. The Probable Origin of SARS-CoV-2 Points to Animals

Phylogenetic analysis of the genome of SARS-CoV-2 shows its affiliation to the species *Severe acute respiratory syndrome-related coronavirus* (SARSr-CoV) of the subgenus *Sarbecovirus* within the *Betacoronavirus* genus [19,20]. Members of this genus were found in African and Asian bats [21,22] as well as Asian pangolins [23]. The closest relative of SARS-CoV-2 was isolated from a horseshoe bat (*Rhinolophus macrotis)* in a Laotian bat cave in 2021 [24]. Although it is possible that spillover occurred through direct bat-to-human contact, the first reported COVID-19 cases in December 2019 were associated with Wuhan wet market activities [25], where live-trapped carnivores, such as raccoon dogs and badgers, were offered for sale, but no bat species [26]. Different animals farmed for food or fur, including civet cats [27], foxes [28], minks [29], and raccoon dogs [30], all proved highly susceptible to sarbecoviruses. Taken together, these circumstances suggest a live intermediate host as the primary source of the SARS-CoV-2 progenitor that humans were repeatedly exposed to, as was the case with the origin of SARS-CoV [31]. Nonetheless, clinical and laboratory confirmations of field infections with SARS-CoV-2-related viruses in such animals before the outbreak of COVID-19 are yet to be made.

In places where environmental and anthropogenic activities predispose to spillover transmissions, the risk of the emergence of a novel zoonotic pathogen is likely to be high [32]. Unfortunately, wild animal hosts and high-risk interfaces facilitating spillover are vastly understudied—and most likely under-reported, in particular in sub-Saharan Africa [33]. Consequently, enhanced monitoring and control efforts beginning from local towards a global strategic one-health surveillance as well as a significant increase in funding for basic and applied research in high-risk countries are being recommended [34,35,36].

### 2.1. Mutations of SARS-CoV-2 Are a Common Feature Facilitating the Crossing of Interspecies Barriers

As with many RNA viruses, the genetic diversity of coronaviruses is due to the high frequency of homologous recombination and accumulation of mutations [37], which supports the breaking of interspecies barriers, varying tissue tropism, and adapting to biological variations [38]. For example, the recombination of the genomes of SARSr-CoV in the spike glycoprotein (S) region is responsible for the mediation of initial cross-species transmission from bats to other mammals [39].

Studies by Lytras et al. (2021) [19] and Temmam et al. (2021) [24] suggested that homologous recombination of a sarbecovirus of unknown origin and a bat coronavirus within the S-region resulted in the progenitor of SARS-CoV-2 [40]. The authors found that the entire genome of SARS-CoV-2 is very similar to SARSr-Ra-BatCoV RaTG13, isolated in 2013 from a horseshoe bat in China, except for the sequence coding for the receptor-binding domain. Viruses genetically related to SARS-CoV-1 were also detected in Ghanaian and Nigerian leaf-nosed bats in 2009 [41] and 2010 [42], respectively, indicating deep-rooted genetic and ecological factors in sub-Saharan Africa yet to be fully explored. Generally, these results point to the complexities of the origin of SARS-CoV-2, in which recombination enabled further evolutionary selection of strains from distinct host species before its spillover to humans. Repetitive close contact between animals and humans, enabling spillover and reverse transmission, may further promote the genetic diversity of coronaviruses [43].

### 2.2. Frequent Transmissions of SARS-CoV-2 to Animals may Create Future Animal Reservoir Hosts

The evolutionary selection of viruses with a greater ability to rapidly adapt to new hosts co-selects for viruses capable of effective intra-species transmission in the new host [17], underpinning disease emergence theory [44,45,46]. Transmission of SARS-CoV-2 from humans to susceptible animal hosts and onwards is likely to amplify mutations that could, in turn, re-infect humans with altered virus variants [47,48], potentially leading to renewed pandemic spread. The susceptibility of animal species depends on several factors, including the compatibility between the viral spike protein and the host receptor ACE2 [49] or alternative receptors such as Neuropilin-1 [50,51] or CD147 [52], the species, and the capacity of the virus to escape the immune system and restriction factors of the new host [47]. For an animal species to act further on as a successful reservoir host, the virus has to become established in the animal population by efficient intra-species transmission, leading to transiently infected animals with prolonged or repetitive phases of shedding infectious particles [53].

Experimental studies suggest that animal species such as cats, dogs [47], ferrets [53], raccoon dogs [54], crab-eating macaques [30], rhesus macaques [55], white-tailed deer [56], rabbits [57], and Syrian hamsters [58] are susceptible to SARS-CoV-2 infection. Moreover, it was found that onward cat-to-cat and ferret-to-ferret [59] as well as hamster-to-hamster transmission [60] can occur through physical or airborne contact.

The worldwide geographical distribution of SARS-CoV-2 in animals according to reports submitted to the OIE is summarized in Figure 1. As of July 2022, 36 countries have reported a total of 679 occurrences of SARS-CoV-2 infections in 24 different animal species of carnivores, primates, ungulates, and rodents [3]. These anecdotally reported cases of natural transmission were mainly directly from infected humans in close contact with animals, particularly to (i) companion animals such as cats and dogs [61] as well as (ii) captive wild animals such as lions, tigers, pumas, snow leopards, and gorillas in zoos [62] and (iii) farmed fur animals such as minks and ferrets. Further on, natural infection was shown in wild white-tailed deer in the U.S. with comparatively high seroprevalences of 37% in the investigated population, implying rapid and efficient spread among this abundant wildlife species [63]. Conversely, animal-to-human transmissions have been demonstrated only in a few reports. Nonetheless, mink-to-human transmission in a mink farm in Denmark [4], the hamster-to-human cluster reported from Hong Kong [64], and the most recently reported case of human infection with a highly divergent SARS-CoV-2 deriving from a wild-tailed deer in Canada [65] demonstrate the potential for further zoonotic transmission cycles accompanied by alterations in nucleotide and amino acid patterns, potentially resulting in new pandemic variants.

However, further investigation is required to determine whether SARS-CoV-2 or other related betacoronaviruses can, in turn, transform new animal hosts into virus reservoirs [66].

## 3. Effective Spillover at the Human–Animal Interface Marks the Entry Point of Novel Infectious Diseases

Spillover events from animal reservoirs into humans are not uncommon in locations with a high frequency of human–animal contact [67]. Serological studies demonstrated evidence of SARSr-CoV-specific antibodies in human residents in rural locations, and even higher rates were recorded in humans living near bat caves in China [68]. Spillover risks increase with human encroachment into rural areas and activities resulting from economic networks around and between rural and urban areas. When a densely packed and immunologically naïve human population in an urban area is then exposed to a novel pathogen, spillover events have a much higher likelihood of resulting in extensive spread [69]. Urbanization characterized by rapid intensification of agriculture, socioeconomic change, and ecological fragmentation can have profound impacts on the epidemiology of infectious diseases [70].

Wildlife populations are heterogeneously distributed, and certain species group in spatial aggregations with livestock and humans, creating interfaces that might be important for the transmission of zoonotic agents [71]. Consequently, anthropogenic pressures can create diverse wildlife–livestock–human interfaces, representing a critical point for cross-species transmission and the emergence of pathogens into new host populations.

Many zoonotic viruses have recently emerged from bats. It has been argued that bats may have an immune system that allows them to coexist with viruses from different virus families [72], hence making them viral reservoirs for filoviruses (e.g., Ebola virus [73], Marburg virus [74]), paramyxoviruses (Hendra henipavirus [75], Nipah henipavirus [76]), rhabdoviruses [77], arenaviruses (*Tacaribe mammarenavirus* has a bat host) [78], and Sarbecoviruses. Spillover to humans happened directly via close contact or bushmeat consumption [79] or indirectly via an intermediate animal species [80]. The complexity of urban systems as networks of physical interfaces across which pathogens can be transmitted between humans and animals exists within the context of societal, cultural, and policy interfaces. Therefore, the investigation of the conditions and the reasons for human–animal contact potentially leading to the emergence of zoonotic diseases requires a multisectoral approach [81,82].

### Spillover Transmission Is Driven by Human Activity

Interspecies transmission of zoonotic pathogens (Figure 2) is mainly driven by human behaviors having a direct or indirect influence on ecosystems and human–animal interactions [83]. Vector-independent transmitted pathogens such as SARS-CoV-2 are distributed through close contact with an infected individual (e.g., droplets, aerosols, fecal-oral) or by contact with contaminated fomites [83].

Direct transmission of zoonotic pathogens applies bi-directionally almost solely regarding the transmission from and to domestic animals in a variety of settings as well as regarding farmed animals or those kept in zoos, where direct contact is inevitable and a substantial part of the human–animal relationship. In the wildlife domain, the direct transmission pathway must be considered one-directional to humans from wildlife hunted and traded for meat consumption or traditional medicine and bi-directional between humans and synanthropic species such as pest rodents [84] or frugivorous bats [85,86]. Other direct contacts between wildlife and humans potentially leading to bi-directional direct transmission may occur during conservation interventions or field research [87]. Remarkably, with the increasing demand for exotic pet species and their trade around the world, the range of pathogens potentially being transmitted within this domain is broadened [88,89].

Fomite transmission from humans to animals pertains whenever waste or sewage can be accessed by domestic or wild animals. For example, SARS-CoV-2 survives on surfaces of personal protective equipment and other household materials for up to three days under normal conditions [90]. Inappropriate discarding or disposal of contaminated personal protective equipment (PPE) such as face masks and gloves as well as tissue wipes, etc., are regarded as a source of COVID-19 infection, especially among stray and wild animals [91]. Even the most elaborate waste management or sewage systems may still be accessed by rodents or birds and therefore cannot prevent onward transmission and spread of pathogens. In the animal-to-human direction, the main transmission pathways to consider are contaminated food of animal origin [92] as well as unwashed manured crops [93]. Furthermore, the use or consumption of contaminated water can be the source of infection with zoonotic pathogens in both directions [94].

## 4. Compounding a Bad Situation: The Impact of Zoonotic Diseases in Africa with a Focus on Nigeria

As experienced with the SARS-CoV-2 pandemic, zoonotic diseases have the potential to threaten human and animal health globally, potentially destabilizing our local economy and impacting food security. In addition, countries of the African continent are still seriously challenged by neglected zoonotic diseases (NZDs), causing huge economic losses and mortality (World Bank, 2018) [95]. The World Health Organization (WHO) has identified eight NZDs: anthrax, bovine tuberculosis, brucellosis, cysticercosis, echinococcosis, leishmaniasis, rabies, and human African trypanosomiasis. These diseases are termed “neglected”, as they mainly affect poor populations who live near domestic or wild animals, often in areas where there are little or no adequate or healthy sanitary conditions [96]. Furthermore, they are neglected due to underestimation of the disease burden, which is usually concentrated in developing countries with ineffective diagnostics and deprived healthcare delivery systems [97]. Nonetheless, NZDs are known to have devastating impacts on human health and welfare, on domestic animal and wildlife health, as well as on risks associated with disease emergence [98]. Additionally, it can be assumed that their presence and persistence in African populations are linked to well-established transmission cycles between humans and animals that may further be used by other or novel zoonotic pathogens with similar transmission modes.

Relatedly, according to a WHO report, while there has been a 63% jump in the number of zoonotic outbreaks in the African region in the decade from 2012–2022 compared to 2001–2011, about 30% of 63% of the substantiated public health events recorded in this region were zoonotic disease outbreaks [99]. The burden of infectious diseases in Africa ranges from newly evolved strains of pathogens (e.g., multi-drug-resistant tuberculosis and chloroquine-resistant malaria) to pathogens that have recently entered human populations for the first time (e.g., HIV-1, Ebola virus, SARS-CoV-2) as well as pathogens that have been historically present in humans but have recently increased in incidence (e.g., Lyme disease, Lassa fever) [100]. With the presence and emergence of many zoonotic diseases, a noticeable trend can be observed. Increasing ecological changes and economical challenges further enhance the development and intensification of factors of (re-)emergence, such as urbanization or socio-cultural behavior such as farming, hunting, and tourism [101].

### 4.1. SARS-CoV-2 Infection Detection Rate in the African Population Is Lower Compared to the Rest of the World

In 2020, the WHO warned in their projection that Africa could likely be the next epicenter of SARS-CoV-2, with an estimated 44 million infections and 190,000 deaths in the first year of the pandemic [102]. Up to July 2022, with 8.7 million cumulated cases and 173,248 deaths reported from the African region, the SARS-CoV-2-attributed mortality in Africa was projected to fall by nearly 94% in 2022 [103]. Based on this record, the enduring mystery of COVID-19 is why the pandemic has not hit low-income African nations as hard as wealthy countries in North America and Europe [104]. The adoption of heightened disease surveillance systems, such as mandatory screening at ports of entry, setting up isolation and quarantine centers, activation of disease surveillance mechanisms from previous influenza and Ebola surveillance systems, as well as contact tracing to swiftly detect and respond to the outbreak, are discussed as potential explanations [105]. However, most African governments are still struggling with strict implementation of containment measures such as border and travel restrictions, bans on large gatherings, social distancing, as well as poor uptake on free vaccinations [106]. Furthermore, the general shortage of diagnostic tests for SARS-CoV-2 on the continent threatens to possibly contribute to a massive underestimation of the true burden of the disease [107].

Concurrently, factors intrinsic to the African population are being discussed to be causative for a lower morbidity and mortality on the continent, such as the general younger demographic of the population, lack of diagnosis in cases of deaths, genetic factors [108], the increased circulation of other coronaviruses having a protective effect against critical COVID-19 [108,109], or the higher burden of other infectious diseases such as malaria leading to an increased alertness of the innate immune system and consequently a lower morbidity and mortality [110].

### 4.2. Socioeconomic and Ecological Factors in Africa Enhance the Probability of Anthropozoonotic Transmission of SARS-CoV-2

Despite the relatively low numbers of SARS-CoV-2 in Africa, particularly in Nigeria, the impending shock on the public health sector could have a devastating impact on the country’s previously strained and fragile health system and could quickly turn into a socioeconomic emergency. The limited availability of diagnostic tests makes the detection of asymptomatic patients nearly impossible and deepens the uncertainty of the potential impact of SARS-CoV-2 infections. Beyond these health and social risks, there are several factors that impact negatively on Nigeria’s economy, including lower trade and foreign investment in the immediate term, falling demands linked with the lockdowns or travel bans, and lastly, a continental supply shock affecting domestic and intra-African trade [111].

Affecting Nigeria’s economic growth, the crisis—together with the effects of climate change—has a significant impact on the overall well-being of people and the number of people living in poverty. According to United Nations (UN) estimates, approximately 30 million more people worldwide could fall into poverty, with a significant rise in the number of acutely food-insecure people [112]. Inadvertently, poverty and food insecurity also have a great impact on human migration, encroachment, as well as changes in human–wildlife interfaces [113,114]. Therefore, policymakers are saddled with the responsibility of enacting measures that will be consistent in tackling realities posed by the SARS-CoV-2 pandemic.

### 4.3. Limited Access to Vaccines Increases the Vulnerability of the Population

The WHO has set a global target of 70% of the population of all countries to be vaccinated by mid-2022 [112]. As of mid-2022, more than 5.33 billion people have received at least one dose of a COVID-19 vaccine globally [115]. However, the vaccination coverage differs strongly in different regions of the globe. In the African region, about 27% of the population was vaccinated with at least one dose [115]. In comparison, in upper-middle- to high-income countries, 81% of the inhabitants have already received at least one dose of vaccine [116]. This inequitable vaccine distribution is not only leaving millions of people vulnerable, but it is also allowing novel, perhaps even more virulent variants of the virus to emerge and subsequently spread across the globe.

As the African region grapples to meet the rising demand for essential vaccination commodities, less than 10% of its nations are projected to hit the year-end target of fully vaccinating 40% of their people by 2022. [117]. The main factors aside from skepticism and hesitancy of some members of the population to be vaccinated [117] are the limited access to vaccines, the disruption of the cold chain, and the shortage in associated medical care [118]. One of the most glaring revelations of the COVID-19 pandemic is the realization that human vaccine manufacturing is almost nonexistent in the entire African continent [118].

### 4.4. Human-to-Animal Transmission May Happen Undetected due to Limited Implementation of Biosecurity and Active Surveillance in Animal Holdings

At global and national levels, veterinary professionals, their representative associations, and animal health regulatory bodies have played various roles in protecting the public’s health during the SARS-CoV-2 pandemic [119]. Controlling highly contagious animal diseases on a large scale has been a recurring challenge for veterinarians for decades [120]. In addition, they understand the epidemiology of the zoonotic disease as well as the risk of potential spillover at the various human–animal interfaces [120].

The Food and Agriculture Organisation of the United Nations (FAO) recommends the systematic and close surveillance of farmed livestock as well as domestic animals, as the transmission to these animals may lead to the further uncontrolled spread of the virus and the potential formation of new reservoirs [121]. In already existing systems in countries in Europe and North America, where the ownership especially of livestock is meticulously registered and monitored by the authorities, the recommended surveillance strategies were rapidly implemented [121]. In countries of Africa without such systems, the implementation of monitoring and control measures is vastly restricted. While SARS-COV-2 transmission from humans to animals was reported in most parts of the world, evidence is scant in Africa, and this may be related to limited surveillance [121].

## 5. Conclusions: Mitigating Future Respiratory Virus Pandemics

In an increasingly globalized world, a spillover of a zoonotic pathogen poses major risks to the global society and economy. This has been well-demonstrated by the ongoing coronavirus disease SARS-CoV-2 pandemic, which has resulted in an unprecedented global public health, social, and economic crisis. Despite our experiences with emerging zoonotic diseases such as SARS, Ebola, Lassa fever, or influenza and subsequently improved national and global surveillance systems, humanity is probably not able to totally prevent the emergence of zoonotic pathogens. Nonetheless, the SARS-CoV-2 pandemic has emphasized the importance of sophisticated global preparedness measures, including monitoring and early detection, prevention of spread, and strategic response mechanisms. Especially in locations where there is a high overlap of wildlife, increasing livestock or agricultural production, and human encroachment, the implementation of early warning systems is needed. Consequently, spotting and mitigating the risks of future spillovers involves working closely with the communities in hotspots for disease emergence and appropriate risk communication. High-risk animal-to-human interfaces must be given special attention. This includes companion animals whose owners have been infected, farmed animals that are often reared in large numbers and are predisposed to viral spread, as well as wildlife in close proximity to human and domestic animal populations. Other high-risk interfaces with human-animal-environmental activities such as land-use change, hunting, and agricultural practices that have been identified to have facilitated viral spillover events should be a focus for education and interventions directed at disease prevention.

Monitoring and control measures limiting zoonotic and onward transmissions should be established and publicly enforced. Particularly in developing countries, farming communities remain vulnerable. Often farms are operated by women and children with limited access to medical, veterinary, and animal production services. The lack of food safety control systems most likely prevents adequate responses to emerging and resurgent zoonotic diseases. Non-pharmaceutical measures, such as physical distancing, strict hand hygiene, respiratory etiquette, appropriate use of facemasks, and home isolation of suspected or confirmed cases as the first line of defense in mitigating potential outbreaks may go along with heavy losses in people’s livelihoods.

Once available, vaccination can provide direct protection in reducing susceptibility among the uninfected and indirect protection in reducing viral spread among the population [122]. Accessibility to an effective vaccine remains a significant challenge in the intervention against SARS-CoV-2. If the effective vaccination rate is satisfactorily high, herd immunity generated by a transmission-blocking vaccine will help to control or even eliminate the spread. Achieving this goal may become more difficult when external constraints affect the deployment of vaccines or when vaccinal and natural immunity is inadequate due to further-evolving immune escape variants.

To successfully tackle zoonotic spillover and its related issues, the resolutions displayed in Table 1 can be considered not just exclusively for the African setting:

## Figures and Tables

**Figure 1 viruses-14-02473-f001:**
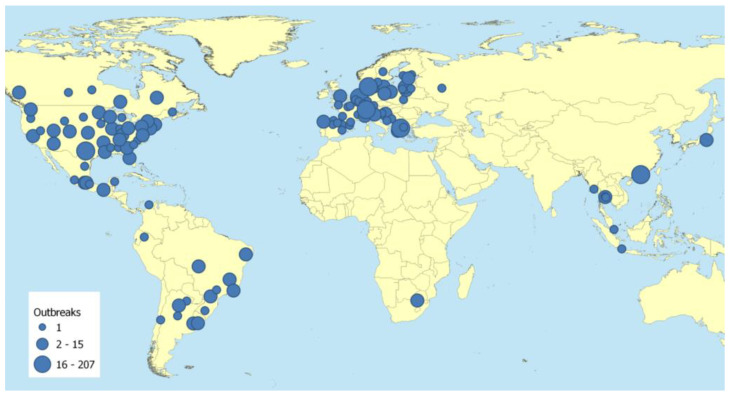
Global distribution of animal infections with SARS-CoV-2. (reprinted with permission from WOAH [3], source: https://www.woah.org/app/uploads/2022/02/sars-cov-2-situation-report-9.pdf, CC BY-NC-ND 4.0 (accessed on 19 November 2021).

**Figure 2 viruses-14-02473-f002:**
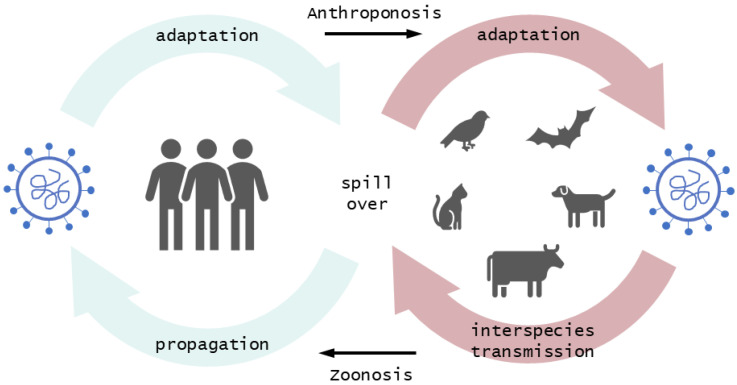
Transmission pathways between humans and animals (image created in Microsoft PowerPoint 2019).

**Table 1 viruses-14-02473-t001:** Resolutions suggested to tackle zoonotic spillover infections and related issues.

Measure	Explanation
Enactments	As a precaution, the public is urged to keep safe distances from wildlife, particularly species that are known to be susceptible to SARS-CoV-2 infections. This can be achieved through enactments such as emergency orders and temporary bans on hunting, trading, and non-essential contact with wildlife. Relatedly, prohibitive measures may be taken to discourage bushmeat hunting and the consumption of raw, unprocessed meat. In all of these, alternative sources of low-cost nutrition should be provided in resource limited communities in order to spare wildlife hunting.
Education	To highlight issues of public health concern, awareness must be raised among the population regarding the possibility of disease transmission from animals to humans and vice versa. While some cultural or religious traditions may predispose to direct contact with wildlife, there is a need to effectively communicate the risks of disease transmission. It may be helpful to recall traditional folk tales that highlight the inherent danger in handling (deceased) wildlife as a learning tool in local public teaching. Many such folklores abound among the native Yorubas of Nigeria and may be applicable in other tribal settings in Africa [123]. Furthermore, early education of pupils on the dangers of emerging or neglected zoonotic diseases and the precautionary measures to be taken can be a significant way to bridge the knowledge gap.
National One Health Strategic Plan	A national one health strategic plan should be actively adopted across the different organizational frontiers (in Africa) to safeguard a holistic impact. From joint (national) research projects to research projects by independent donors, combined efforts must achieve the objective of containing virus spread. The capabilities of self-determined and independent, top-level research should be continuously increased via access to funding, training, and education. This enables local and independent monitoring and rapid identification of novel emerging pathogens.
Vaccine equity	The autonomous development and production of vaccines in Africa must be actively supported. This will create the enabling conditions to achieve a rapid response to emerging or pandemic pathogens and increase the global vaccination coverage while continually developing vaccines for mutant strains of the virus [117].
Heightened Monitoring	Gathering data on human practices as well as contact with animals in settings with diverse host assemblages will ensure effective analysis regarding potential spillover risks. Environmental, veterinary and medical scientists may further investigate the pathogens present and exchanged at the identified human–animal interfaces of risk. This will give an informed direction towards critical points for disease control and behavior change interventions aimed at prevention.

## Data Availability

Not applicable.

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
