# Peer review of "SARS-CoV-2 at the Human–Animal Interface: Implication for Global Public Health from an African Perspective"

_viruses, 2022, doi:10.3390/v14112473_

Round 1

Reviewer 1 Report

Ebere Roseann Agusi et al. has wonderfully explained the cause of zoonotic origin and tried to correlate it with African context, especially why this country could not get highest hit compared to wealthy counties. 

Also, authors have rightly highlighted that "One of the most glaring revelations 276 of the COVID-19 pandemic is the realisation that human vaccine manufacturing is almost nonexistent in the entire African continent [115]." and "This inequitable vaccine distribution is not only leaving 269 billions of people vulnerable, but it is also allowing novel, perhaps even more virulent variants of the virus to emerge, and subsequently spread across the globe".

Manuscript and the work look impressive to me. However, I have a few minor suggestions for the authors to improve the MS as follows: 

1. "global One Public 20 Health impact" in the abstract does not look good neither grammatically nor linguistically. Please use alternate words. Similar is the case for "One Health surveillance".

2. Put reference 63 correctly in parentheses in line number 134.

3. Image 2 was created in BioRender. Did you got license fort he same? Free version does not allow you to publish any figure in any journal.

4. Please use a short and crisp title for the article. Also, improve the language of the abstract carefully. Use direct sentences.

5. It will be good to add a section "genetic factors in African population as a plausible cause for lower SARS-CoV-2 infection rate compared to rest of the world". Add a table for such data.

6. Also, please introduce one or two tables to improve the reading of the MS. Only text may not attract many reader and the understanding may not be that easy.

Author Response

Response to Reviewer 1 comments:

(For easier reading, we added our responses in blue after the reviewer's comments)

Ebere Roseann Agusi et al. has wonderfully explained the cause of zoonotic origin and tried to correlate it with African context, especially why this country could not get highest hit compared to wealthy counties.

Also, authors have rightly highlighted that "One of the most glaring revelations 276 of the COVID-19 pandemic is the realisation that human vaccine manufacturing is almost nonexistent in the entire African continent [115]." and "This inequitable vaccine distribution is not only leaving 269 billions of people vulnerable, but it is also allowing novel, perhaps even more virulent variants of the virus to emerge, and subsequently spread across the globe".

Manuscript and the work look impressive to me. However, I have a few minor suggestions for the authors to improve the MS as follows:

  1. "global One Public 20 Health impact" in the abstract does not look good neither grammatically nor linguistically. Please use alternate words. Similar is the case for "One Health surveillance".

The wordings in line 24 and in line 30 were amended as recommended.

  1. Put reference 63 correctly in parentheses in line number 134.

Done.

  1. Image 2 was created in BioRender. Did you got license for the same? Free version does not allow you to publish any figure in any journal.

The image was recreated in Microsoft PowerPoint, using icons and pictograms that are license-free.

  1. Please use a short and crisp title for the article. Also, improve the language of the abstract carefully. Use direct sentences.

The title was shortened to “SARS-CoV-2 at the human-animal interface: Implication for global public health from an African perspective” as recommended. The language of the abstract was carefully revised according to the reviewer’s recommendations.

  1. It will be good to add a section "genetic factors in African population as a plausible cause for lower SARS-CoV-2 infection rate compared to rest of the world". Add a table for such data.

In lines 242ff, a new section was inserted, in which the suggested points are addressed and discussed. While the data on genetic factors in African populations are certainly very interesting, the focus of our study is on spill-over events at the human-animal interface, which fall within our field of expertise, while intrinsic human factors do not. Therefore, we decided to keep it brief and hope you accept that we do not go into further detail here.

  1. Also, please introduce one or two tables to improve the reading of the MS. Only text may not attract many reader and the understanding may not be that easy.

A table was added in section 5. for easier readability.

Reviewer 2 Report

Overall, good work compiling the different sources of information into this review.

1.       The author writes, “SARS-CoV-2 offer great potential for a future reservoir of pandemic viruses evolving from the current SARS-CoV2” Can you elaborate on this mechanism, how SARS-CoV-2 can have great potential for future pandemics?

2.       Do you also think the drop in vaccination for other infectious diseases in the last few years can potentially affect the spread of viral infections?

3.       “The susceptibility of animal species depends on several factors, including the compatibility between the viral spike protein and the host receptor ACE2 of the species, and the capacity of the virus to escape the immune system and restriction factors of the new host.” How about the other receptors that SARS-CoV-2 is probably using? Does that matter in the compatibility of spread?

Author Response

Response to Reviewer 2 comments:

(For easier reading, we added our responses in blue after the reviewer's comments)

Overall, good work compiling the different sources of information into this review.

  1. The author writes, “SARS-CoV-2 offer great potential for a future reservoir of pandemic viruses evolving from the current SARS-CoV2” Can you elaborate on this mechanism, how SARS-CoV-2 can have great potential for future pandemics?

The great potential of SARS-CoV-2 for future pandemics is the possible alteration of the virus in animal reservoirs, from which new variants with altered characteristics could then spill-over to the human population again, leading to a renewed pandemic spread. This is further elaborated in lines 109ff and lines 142ff.

  1. Do you also think the drop in vaccination for other infectious diseases in the last few years can potentially affect the spread of viral infections?

This is an interesting point. From our current point of knowledge and literature research, there is no indication of a correlation of e.g. the drop in Polio vaccination and the spread of viral infections (other than Polio) in general. But this would need to be further investigated in a meta-analysis, leading to a whole new review.

  1. “The susceptibility of animal species depends on several factors, including the compatibility between the viral spike protein and the host receptor ACE2 of the species, and the capacity of the virus to escape the immune system and restriction factors of the new host.” How about the other receptors that SARS-CoV-2 is probably using? Does that matter in the compatibility of spread?

Thank you! Good point! ACE2 was highlighted here example wise, as the efficiency of ACE2 usage was identified as a key determinant of SARS-CoV-2 transmissibility, and was being investigated best with regard to susceptibility of animals. Other receptors, like NRP-1 or CD147, have an influence on the capability of spread, too. We amended the statement in lines 121ff, and added 2 references.